# Supporting the Wound Healing Process—Curcumin, Resveratrol and Baicalin in In Vitro Wound Healing Studies

**DOI:** 10.3390/ph16010082

**Published:** 2023-01-06

**Authors:** Kacper Jagiełło, Oliwia Uchańska, Konrad Matyja, Mateusz Jackowski, Benita Wiatrak, Paweł Kubasiewicz-Ross, Ewa Karuga-Kuźniewska

**Affiliations:** 1Department of Process Engineering and Technology of Polymer and Carbon Materials, Faculty of Chemistry, Wroclaw University of Science and Technology, ul. Gdańska 7/9, 50-344 Wrocław, Poland; 2Department of Reproduction and Clinic of Farm Animals, Faculty of Veterinary Medicine, Wrocław University of Environmental and Life Sciences, pl. Grunwaldzki 49, 50-366 Wrocław, Poland; 3Department of Micro, Nano, and Bioprocess Engineering, Faculty of Chemistry, Wroclaw University of Science and Technology, ul. Norwida 4/6, 50-373 Wrocław, Poland; 4Department of Pharmacology, Wroclaw Medical University, ul. Jana Mikulicza-Radeckiego 2, 50-345 Wrocław, Poland; 5Department and Clinic of Dental Surgery, Wroclaw Medical University, ul. Krakowska 26, 50-425 Wrocław, Poland; 6Department of Epizootiology with Exotic Animal and Bird Clinic, Faculty of Veterinary Medicine, Wrocław University of Environmental and Life Sciences, pl. Grunwaldzki 45, 50-366 Wrocław, Poland

**Keywords:** bioflavonoids, difficult-to-heal wounds, Brian and Cousens model, scratch assay, MTT assay

## Abstract

The purpose of the investigation was to evaluate the effect of the selected bioflavonoids curcumin, resveratrol and baicalin on the wound healing process in an in vitro model. In the study, Balb3t3 and L929 cell lines were used. The first step was the evaluation of the cytotoxicity of the substances tested (MTT assay). Then, using the scratch test (ST), the influence of bioflavonoids on the healing process was evaluated in an in vitro model. The second stage of the work was a mathematical analysis of the results obtained. On the basis of experimental data, the parameters of the Brian and Cousens model were determined in order to determine the maximum value of the cellular and metabolic response that occurs for the examined range of concentrations of selected bioflavonoids. In the MTT assays, no cytotoxic effect of curcumin, resveratrol and baicalin was observed in selected concentrations, while in the ST tests for selected substances, a stimulatory effect was observed on the cell division rate regardless of the cell lines tested. The results obtained encourage further research on the use of substances of natural origin to support the wound healing process.

## 1. Introduction

Difficult-to-heal wounds are one of the most serious problems in the daily practice of veterinarians. Healing processes are generally effective in healthy animals. The organism reacts to skin damage by initiating numerous complex repair processes. Depending on the scale and type of defect, the healing process takes between a few days and several weeks. The effect of healing is to close the skin loss by filling it with scar tissue. Then, the scar rebuilds, which can take weeks, months, or even years [1,2,3,4].

The wound becomes clinically difficult in the event of a problem related to the defect itself, such as infection, damage to deep tissue structures or the nature of the damage (e.g., mechanical, chemical, radiation) [1]. The mentioned conditions have a significant influence on the speed and quality of the healing process. Geriatric patients, who suffer from chronic diseases and metabolic disorders, experience a much more difficult wound healing process [2]. The time it takes to close the wound can take months or even years, and the wound often renews itself [3,4].

The basis for the treatment of chronic skin defects is a holistic approach, i.e., the treatment of the general condition of the patient and local wound therapy [2,5]. Local wound treatment consists of selecting the appropriate dressing material. The dressing is not only to protect the wound surface but also to actively support the healing process [6]. A suitable moist environment is ensured by the use of hydrogel dressings that support cell proliferation and debridement of non-viable tissue. Most wound dressings work by providing moisture to the wound or keeping the moisture in the wound bed for a longer period of time. For example, dressings that use hydrofiber technology such as AQUACEL, which further supports debridement, provides a moist environment and absorbs excessive exudate [7]. There are various medical dressings available on the market for wound treatment. Their healing activity often depends on the additives used, e.g., different kinds of dressings containing silver ions, which strongly support the antibacterial effect. Research has shown that nanocomposite hydrogels used for wound treatment exhibited significant inhibition against Gram-positive and Gram-negative bacteria due to Ag^+^ ions released from silver nitrate [8,9,10]. Another readily used wound healing material is, for example, honey extract, which improves wound removal and cell proliferation, leading to faster closure of wound edges [11,12,13,14,15]. It is also beneficial due to its high sugar content, which creates a higher osmotic gradient that pulls fluid through the subdermal tissue and provides an additional source of glucose for the growth of cellular components in the wounded area [11]. The use of growth factors is also observed in dressings due to their ability to activate fibroblasts, increase their proliferation and, as in the case of the aforementioned substances, accelerate the healing of skin loss [16,17,18]. For years, scientists have focused their attention on the potential of substances of plant origin [19,20,21]. Among this group, flavonoids deserve special attention, of which the following substances caught our attention: curcumin, resveratrol and baicalin.

Curcumin is a substance extracted from the *Curcuma longa* rhizome, a member of the ginger family *Zingiberaceae*, whose healing properties have been used in Chinese and Indian natural medicine for centuries [9]. Curcumin has properties that regulate the intensity of the inflammatory process, one of the most important being its ability to reduce the production of cytokines released by monocytes and macrophages, which play a major role in the enhancement of the inflammatory response of the body [13]. This is the body’s first reaction in response to the factor that causes the wound. The persistent inflammatory process prevents wound granulation and therefore has the ability to regulate it, making curcumin a valuable substance important in the healing process [11]. Furthermore, curcumin also has antioxidant properties and stimulates angiogenesis and cell proliferation processes, so it is used particularly frequently and easily to produce therapeutic dressings for wounds that arise in the course of diabetes [12,13].

Resveratrol is a natural polyphenol present in the highest amounts on the skin of grapes and red wine. It has a strong antioxidant, anti-inflammatory, antifungal and antibacterial effect as well as cardio- and neuroprotective properties [14]. Numerous studies also show its effectiveness against skin, prostate, lung, colon and liver cancer cells [15,16]. Furthermore, resveratrol has the ability to induce cell proliferation and accelerate wound closure in in vivo models in studies conducted on rats [21]. In vitro studies have shown protective effects on fibroblasts by reducing oxidative stress. Due to its antioxidant properties, resveratrol can stabilize cell proliferation, improve migration quality and ultrastructural behavior [18] Furthermore, in vivo results revealed significantly improved wound healing after resveratrol treatment based on antioxidant activity [17]. Angiogenesis also plays a key role in wound healing, and resveratrol is discussed as an important factor that influences neovascularization [19]. This process is essential for tissue regeneration and is mainly regulated by vascular endothelial growth factor (VEGF), whose expression can be significantly increased by resveratrol [20]. The effect of resveratrol on the production of pro-inflammatory cytokines, which are released in large amounts during the inflammation that accompanies wound formation and healing, was also investigated. The presence of cytokines such as interleukin (IL)-1β, IL-6, C-reactive protein (CRP) and TNF-α2 prolongs the inflammatory phase and delays the healing process. Studies have shown that patients who received resveratrol during treatment had lower levels of these mediators and their wounds healed faster [21,22]. Therefore, it seems promising to carry out more studies on the possibilities of wound therapy using an external application of resveratrol due to its multifaceted effect.

Baicalin is a flavonoid extracted from the roots of *Scutellaria baicalensis*, a plant that is used in traditional Chinese medicine. It is widely known for its anticancer, antimicrobial and antioxidant properties [23,24,25]. Studies also show its potential use as an angiogenesis-regulating agent, which can be successfully used in the wound environment for tissue regeneration and vascularization [24] Research conducted on the wound healing process in immunocompromised dogs has shown that they have a slower healing ability compared to their non-immunocompromised counterparts. Digital analysis data showed that wounds treated with a dressing containing baicalin, β-sitosterol and berberine as active ingredients are characterized by an enhanced epithelialization process, faster contraction and smaller wound area compared to honey-treated wounds alone [26]. Baicalin also has the ability to reduce serum blood glucose levels and oxidative stress markers in rats with experimentally induced *diabetes mellitus*. The study showed that baicalin upregulated the expression of, e.g., VEGF-c, TGF-β, Tie-2 and SMAD2/3, implicating its potential antidiabetic and wound healing properties. Thus, baicalin can be emphasized as a potential candidate for the treatment of chronic wounds [27].

In this study, the effects of selected bioflavonoids on the in vitro wound healing process were examined. The influence of different concentrations of curcumin, resveratrol and baicalin on cell viability (MTT Assay) and cell migration (scratch assay) was evaluated. The use of mathematical modeling is designed to determine the most effective concentrations of selected substances that support the fibroblast proliferation process. So far, no studies have been published that would use the Brian and Cousens model to describe the mitochondrial or cellular response to applied doses of bioflavonoids. The results obtained can be helpful in developing a modern healing dressing based on natural medicinal products.

The cell lines used in the study are well characterized. Furthermore, fibroblasts are one of the foundations of the healing process. These cells multiply and create a matrix for the developing tissue. It is the fibroblasts that are involved in scar formation. It seems natural to use them in the study of substances that are supposed to have a wound-healing effect. Supporting the proliferation of fibroblasts in the wound has an impact on the improvement in the healing process.

## 2. Results and Discussion

### 2.1. Model Fit and Statistical Analysis for MTT Test Results

The results of the MTT are presented in Figure 1 respectively. The values of the estimated parameters of the Brian and Cousens model for MTT results can be found in Table 1. The graphs show the prediction of the model and the maximum of the function corresponding to the highest cell viability. Only the mean values of the cell responses obtained are shown to maintain the readability of the graph.

It should be noted that low values of R^2^ (Table 1) indicate a small difference between the mean value of the data and the prediction of the model. A low value of RMSE indicates a good fit of the model.

The *t* test was performed to indicate statistical differences between the mean values of the samples, which were suspected to show the stimulation effect and the mean in the control. One of the assumptions of *t* test is that both samples are approximately normally distributed. The normality of the distribution was analyzed using the Anderson–Darling test, and the results are available in the Appendix A. The hypothesis that the data come from a population with a normal distribution was not rejected for most of the samples tested at α = 0.025. The null hypothesis was rejected only for samples obtained from L929 cell line with 3 mg/L curcumin and 0.5 mg/L baicalin at α = 0.001. Therefore, these two samples were not analyzed with the *t* test. The results of the *t* tests, obtained for the experimental points of the MTT tests, are presented in Table 2.

### 2.2. Model Fit for Scratch Assay Tests

After confirmation of the absence of cytotoxicity with the MTT tests, it is possible to proceed to the study of the pro-proliferative effect of selected bioflavonoids with the ST test. The results of the experiments are shown in Figure 2, while the parameters of the Brian and Cousens model are presented in Table 3.

## 3. Discussion

### 3.1. Model Fit and Statistical Analysis for MTT Test Results

Based on the results of the MTT tests, it can be concluded that none of the substances tested showed a cytotoxic effect on the Balb3t3 and L929 fibroblast cell lines in the concentration range of 0 to 20 mg/L (Figure 1 and Table 1). The greatest negative effect on viability was observed for the L929 cell line for a curcumin concentration of 25 mg/L. Fibroblast cell viability for this concentration was 67.15 ± 5.01%. According to the ISO 10993-5 standard, a substance in a given concentration should be considered to have a cytotoxic potential if cell viability is <70% [28].

Due to the Brian and Cousens model, the half-maximum effective concentrations (EC50) were determined for selected bioflavonoids. The highest values in both lines were obtained for resveratrol; they were an order of magnitude higher than for curcumin and baicalin. It shows a slight influence of resveratrol on fibroblast mitochondrial activity. Parameter *b* was in a similar range of values for the bioflavonoids used regardless of the cell line used, excluding the combination of resveratrol and L929, where it was highest (3.53), indicating the steepest slope of the model curve.

The lowest values of *f* were determined for resveratrol. Furthermore, *E_max_* values in the range of 1.00 suggest that this particular bioflavonoid does not have a stimulating effect on mitochondrial activity in L929 and Balb3t3 cell lines in tested time intervals. For curcumin and baicalin, the *E_max_* values did not exceed 1.35. It may suggest that these substances have a relatively low stimulation effect on fibroblast cell activity after 24 h of exposure. The *f* parameter could not directly provide information about the *E_max_* but it should be read with calibrated *x_op_*. The low parameter *f* and the low *x_op_* for resveratrol in both cell lines allowed *E_max_* values greater than 1.00, but, on the other hand, the relatively low *f* and the high *x_op_* for L929 curcumin resulted in the value of *Emax* at the highest level of 1.35.

Increased mitochondrial activity can be correlated with growth rate. The higher the growth rate, the higher the mitochondrial activity. If cell divisions occur more rapidly in the first hours of incubation, it is possible that they were associated with much higher Emax values than those recorded after 24 h.

The observed cytotoxic effect is consistent with the observations of other researchers, e.g., Alqahtani et al. [29]. In the range of curcumin concentrations tested in the MTT assay (2–64 μM) on the human keratinocyte line (HuCaT), the cytotoxic effect was detected for a curcumin concentration greater than 32 μM. Our studies carried out in the mouse fibroblast lines Balb3t3 and L292 cover the concentration range of up to 25 μg/mL (about 67.86 μM) and at the highest concentration of this bioflavonoid, a decrease in cell viability could be observed. Another study carried out in human gingival fibroblasts (HGF) also shows that 0.1–20 μM curcumin solutions did not decrease cell viability, while solutions greater than 30 to 50 μM showed a cytotoxicity effect related to increasing concentration [30,31].

Researchers Lantto et al. [32] did not observe any resveratrol cytotoxicity in the range of 100 µM (22.8 mg/L) for SH-SY5Y (neuroblastoma cells) and CV1-P (fibroblasts) cell lines. A cytotoxic effect was observed for the CV1-P line at concentrations of 300 µM (68.5 mg/L) and 400 µM (91.3 mg/L). Similar studies were also carried out for curcumin, but here the cytotoxic effect was observed at concentrations greater than 25 µM (9.2 mg/L) for SH-SY5Y cell lines, while for the CV1-P line the cytotoxic effect was not observed at concentrations greater than 100 µM (36.8 mg/L). No cytotoxic effect was also observed in research on human dermal fibroblast cells in resveratrol concentrations in the range of 5.55 µM to 150 µM (1.3 to 34.2 mg/L) [33].

Experiments on the effect of baicalin on 3T3 fibroblasts carried out by Manconi et al. [34] also did not show a cytotoxic effect on baicalin concentration in PBS in the concentration range 0.1–20 mg/L. Experiments in rat lung fibroblasts also confirmed that there was no cytotoxic effect in mass concentrations of 20 to 80 mg/L with an addition of 20 mg/L of bleomycin. No cytotoxic effect was also observed in nasal fibroblasts up to 50 µM (22.3 mg/L) of baicalin [35,36].

The observed effect could be explained by a higher sensitivity of cancer cells compared to fibroblast cells to natural phenolic substances [32]. Our research seems to support reports that natural bioflavonoids do not have cytotoxic efficacy on fibroblast cells in the concentrations examined.

Based on the *t* tests (Table 2), statistically significant differences indicating the stimulating effect on cell viability were observed only for Balb3t3 cell line samples with 1 mg/L baicalin and L929 cell line samples with 6 mg/L curcumin. Although the administered MTT test could not exclude the stimulating effect of curcumin, resveratrol and baicalin on the L929 and Balb3t3 fibroblast cell lines. To fully understand the impact of the examined bioflavonoids on mitochondrial activity, different exposition times should be examined. However, our research appears to support reports that natural bioflavonoids do not have cytotoxic efficacy on fibroblast cells at the concentrations examined.

### 3.2. Model Fit for Scratch Assay Tests

Regardless of the cell lines tested, a stimulating effect of selected bioflavonoids was observed on the rate of cell division (Figure 2 and Table 3). It should be noted that the ST test provides information on the cumulative increase in the amount of new cells during the incubation period, while the MTT test provides information on the approximate instantaneous (2 h) mitochondrial activity.

In the case of curcumin, the maximum cell division rate for the concentration of 4.73 mg/L is approximately 4.5 times higher than in samples in which no bioflavonoid was added for the Balb3t3 line and more than three times higher than in the control sample for the concentration of 5.15 mg/L for the L929 line.

In both cell lines, a stimulating effect of resveratrol-supplied medium was observed in the range of 0.5–4 mg/L. The maximum cell division rate determined for resveratrol according to the calculated model is more than 3.5 times higher for the Balb3t3 line at 1.01 mg/L and approximately 5 times higher for the L929 line at 1.29 mg/L than for controls. At higher concentrations, a negative effect of resveratrol can be observed on the ability of fibroblasts to divide.

The maximum cell division rate observed for baicalin is 6 times higher for the Balb3t3 line for a concentration of 2.28 mg/L and more than 5.4 times for the L929 line for a concentration of 2.57 mg/L than for the relevant control samples. Unlike the remaining bioflavonoids for higher concentrations of baicalin (above 12 mg/L), a 2.5-fold stimulation of fibroblasts was still observed. This may indicate low toxicity of a high concentration of baicalin in the tested fibroblast lines.

The Brian and Cousens model was successfully fitted to all combinations of bioflavonoid cell lines for ST. The highest effect of the compound tested was observed for the combination of baicalin—Balb3t3 and the lowest for curcumin—L929. The coefficient of determination (R^2^) did not drop below 0.74, while the highest RMSE value determined was 0.78. The corresponding slope values (b) and inflection point values (e) for the same substance differ for different cell lines by a maximum of 15% and 30%, respectively. For the same bioflavonoid, the models show the maximum stimulation of cell division at concentrations of the test substance that differ by a maximum of 0.4 mg/L. In all cases, the x_op_ was greater for L929 fibroblasts. This may indicate a very similar mode of action of the bioflavonoid in cells regardless of the type of fibroblast lineage.

For curcumin, Alqahtani et al. observed an increased proliferation effect in the L929 fibroblast cell line [29]. Studies suggest that curcumin-loaded lignin nanoparticles have a stronger stimulative effect than 50 µM (18.4 mg/L). Mirzahosseinipour et al. also observed the wound healing and antimicrobial properties of 50 mg/L curcumin in HDF fibroblast cells [37]. In this case, the form that delivers curcumin encapsulated in silica nanoparticles was also more effective than that of the bioflavonoid solution. Similar to the previous study, researchers did not present results for other concentrations of bioflavonoid solutions.

The process of cutaneous wound healing is extremely complex and depends on an intricate interplay between several highly regulated factors that work together to restore injured skin to repaired barrier function. The ideal dressing whose application would support and accelerate wound healing is faced with more and more challenges. Numerous studies are being conducted on various auxiliary substances and their applications that would meet all requirements, including protection against bacterial infection [38,39,40], reduction of the inflammatory process [41,42] and acceleration of cell proliferation aimed at reconstructing damaged tissues [43,44].

The substances selected for this study, curcumin, resveratrol and baicalin, have already been shown to have positive effects on wound healing [13,17,19,25,45]. Many studies have shown in which amount and application forms exhibit the most effective supportive effects [5,14,19,20,23,24,25,28,35,36,37,38,39,40,41,46] Understanding the optimal dose of these substances is essential for multiple goals and, above all, their complex role in the inflammatory response in wound repair needs to be addressed before further clinical development.

## 4. Materials and Methods

### 4.1. Bioflavonoids

In the presented study, the following bioflavonoids were used: curcumin, resveratrol and baicalin. All substances were sourced from Sigma-Aldrich (St. Louis, MO, USA).

### 4.2. Cell Lines

The study presented was carried out on L929 and Balb3t3 cell lines. Cell lines were obtained from Sigma-Aldrich (St. Louis, MO, USA). Both cell lines were cultured at 37 °C, 5% CO_2_ and the relative humidity was set at 95%. Cell lines were passed twice a week and experiments were performed on cells between passages 15 and 25. The culture medium was changed every 48 h. Each dictation was performed according to the presented procedure: after removing the supernatant from L929 and Balb3t3 cells, they were washed with PBS (phosphate buffered saline), then treated with TrypLE Express solution (Gibco, Thermo Fisher Scientific, Waltham, MA, USA) and incubated for 5 min at 37 °C. Next, the cells were transferred to a centrifuge tube, complete medium (to inactivate TrypLE Express) was added (to inactivate the TrypLE Express), and cells were centrifuged for 5 min at 1000× *g*. The supernatant was removed and cells were resuspended in fresh medium. The viability of a cell line was checked every day by microscopy observation of the cell culture morphology.

#### 4.2.1. L929

Mice fibroblast L929 cells (subclone of parental strain L, strain L was derived from normal subcutaneous areolar and adipose tissue of a 100 day old male C3H/An mouse) were obtained from Sigma-Aldrich (St. Louis, MO, USA) and cultured in Minimum Essential Medium Eagle (EMEM).

#### 4.2.2. Balb3t3

Mice fibroblast Balb/3T3 cells (clone A31 from a mouse embryo donor) were obtained from Sigma-Aldrich (St. Louis, MO, USA) and cultured in Minimum Essential Medium (MEM).

### 4.3. Cell Viability

The MTT-3-(4,5-dimethylthiazol-2-yl)-2,5-diphenyltetrazolium bromide assay (Sigma-Aldrich, St. Louis, MO, USA) was used to evaluate the effect of the bioflavonoids chosen on cell viability according to the ISO 10993 standard, part V [28]. The MTT assay was performed on commercial L929 and Balb3t3 cell lines.

The test stocks were prepared by the following procedure: to the bioflavonoids, DMSO (dimethyl sulfoxide) was added to the final concentration set at 10 mM. The stock solutions were stored at −80 °C paying attention that the storage time does not exceed 6 months. The stock solutions were dissolved every time before the planned biological assay with appropriate culture media. The culture medium used was reduced to the FBS level, up to 5%. During the test, it was crucial to set the DMSO concentration level lower than 1% in the highest bioflavonoid concentration of the test.

Each cell line was seeded in 96-well plates at a density of 1 × 10^4^ cells/mL per well. The plates were incubated for 24 h to allow the cells to attach to the bottom of the plate. After this time, each line, in addition to the control, was exposed to different concentrations of bioflavonoids tested. The concentrations examined were: 0.5, 1, 3, 6, 9, 12, 15, 20, 25 mg/L. The exposure time to the test substances was set at 24 h. At the end of the exposition time, the medium was removed and 50 µL of 1 mg/mL of MTT solution was added to each well and then incubated at 37 °C for another 2 h. At this time, crystals of formazan were created. Finally, the supernatant was removed and the formazan crystals formed were dissolved in 100 µL of isopropanol (30 min incubation) for spectrophotometric evaluation. Absorbance was measured with a Victor2 microplate reader (PerkinElmer, Waltham, MA, USA) at 570 nm. The formazan concentration in each sample was expressed as a relative value of the mean formazan concentration in the control and used in further analysis. Rav results can be find in the Appendix A.

### 4.4. Scratch Assay

A scratch test (ST) was used to perform a wound healing evaluation under in vitro conditions. This procedure allows us to study the collective migration of cells in a two-dimensional confluent monolayer under test conditions [47]. ST was performed as follows:

Cell lines were seeded in 6-well culture plates. The cell was incubated at 37 °C, 5% CO_2_, relative humidity 95% for 24 to create a confluent monolayer. After the cell monolayer was achieved, the scratch-making procedure was performed. A sterile plastic micropipette tip was used to simulate an in vivo wound by creating a cell-free zone with a straight edge across the cell monolayer in each well. A gap width of 0.7 mm allows observation at 4 or 10 magnifications. After the scratches were made, the wells were washed with basal medium to remove cell debris and a complete medium was added. Test bioflavonoids at the following concentrations were added to the appropriate wells: 0.5, 1, 3, 6, 9, 12, 20 mg/L. The control sample was not exposed to bioflavonoids. Finally, the tested systems were incubated for 24 h under the following conditions: 37 °C, 5% CO_2_, relative humidity 95%. After incubation, each wound was scored at 4 and 10× magnification. The width of the lesion was measured in 3 places. Each defect was measured two times: T0, at the time of wound execution and T24, after 24 h of incubation. The mean rate of the cell gap bridge was determined over the course of 24 h. The results were expressed as relative values of the bridging rate in the control sample. Rav results can be find in the Appendix A.

### 4.5. Model Fit and Statistical Analysis

The Brian and Cousens model was used to analyze the influence of bioflavonoids on cell viability and cell migration [46,48]. The model can be used to describe the effects of inhibition and stimulation.
(1)$$E=c+\frac{d-c+fx}{1+\exp\left(b\left(ln\left(x\right)-ln\left(e\right)\right)\right)}$$
where E—relative effect; x—a concentration of bioflavonoid [mg/L]; e—inflection point equal to EC50 for *c* = 0 and *d* = 1 [mg/L]; b—parameter which determines the slope; and 𝑐 and 𝑑—limit at *x*→0 and limit at *x*→∞ equal to 0 and 1, respectively; f—stimulation parameter.

The values of the parameters of the Brian and Cousens model were estimated using the nonlinear least squares method in MATLAB. To obtain a more reliable model fit, some additional constraints were provided: *b* > 1 and *f* > 0 for all cases and *e* > 1 for the MTT assay for cell lines L929 and baicalin. For every estimated parameter, the 95% non-simultaneous prediction bounds for new observations were obtained. The goodness of fits was characterized by the coefficient of determination (R^2^) and the root mean squared error (RMSE).

The Nelder–Mead Simplex Method implemented in MATLAB was used to find the concentration of bioflavonoids $x_{op}$ for which maximum value of the relative effect $E_{max}$ is reached by maximizing the value of the Brian and Cousens model (1) for each set of estimated parameters. 

Based on determined $E_{max}$, the experimental data sets obtained for the concentration of bioflavonoids next to $x_{op}$ were chosen for further statistical analysis. The Anderson–Darling test was performed to check whether the experimental data are normally distributed at significance levels of 5%, 2.5% and 1%. The two-sample *t* test was performed to indicate the statistical difference between the mean values of the relative effect of the experimental data chosen and the control samples at the 5% significance level [49]. Both the Anderson–Darling and the *t* test were performed in MATLAB. Anderson – Darling test results and *t*-test results for all bioflavonoids can be found in the Appendix A.

## 5. Conclusions

This article examined the cytotoxicity of natural bioflavonoids and their effect on mitochondrial and proliferation activity in L929 and Balb3t3 cell lines. The experiments confirmed the lack of cytotoxicity of the selected substances in the concentration range tested in the Balb3t3 and L929 fibroblast lines. The selected bioflavonoids curcumin, resveratrol and baicalin showed a simulative effect on the proliferative capacity of both cell lines. The Brian and Cousens model could be used to describe a stimulative effect of selected bioflavonoids. As a result of the obtained results, all of the examined substances raise hopes that they may be used in difficult-to-heal wound treatment.

## Figures and Tables

**Figure 1 pharmaceuticals-16-00082-f001:**
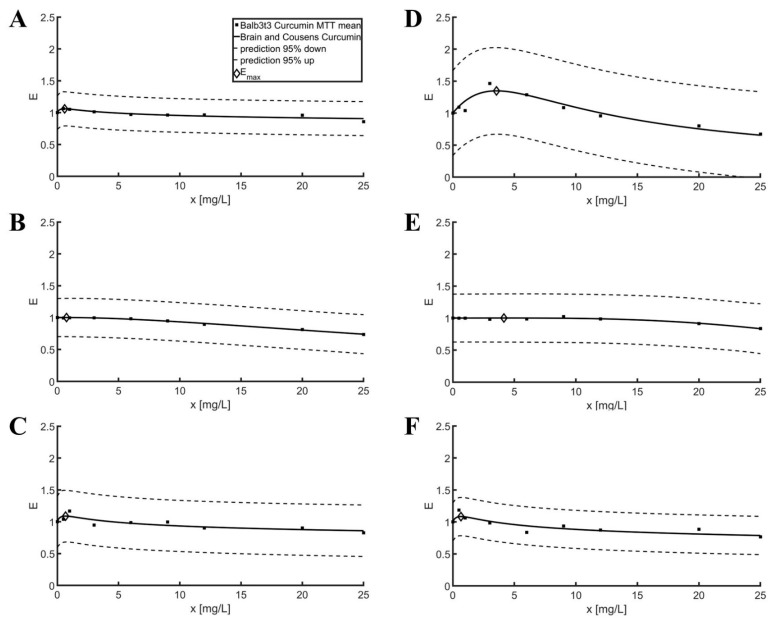
Brian and Cousens model curve fitting with 95% prediction bounds for the MTT assay. (**A**) Curcumin—Balb3t3; (**B**) Resveratrol—Balb3t3; (**C**) Baicalin—Balb3t3; (**D**) Curcumin—L929; (**E**) Resveratrol—L929; (**F**) Baicalin—L929.

**Figure 2 pharmaceuticals-16-00082-f002:**
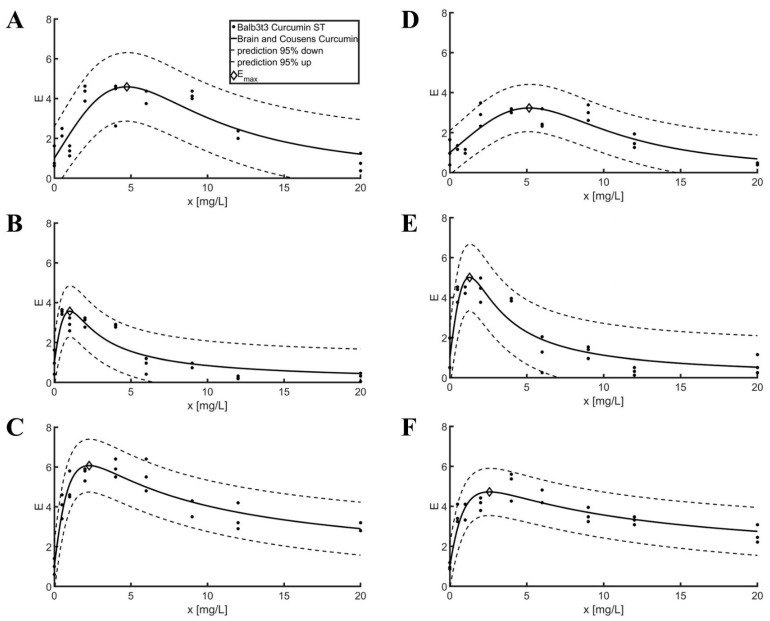
Brian and Cousens model curve adjustment for ST test. (**A**) Curcumin—Balb3t3; (**B**) Resveratrol—Balb3t3; (**C**) Baicalin—Balb3t3; (**D**) Curcumin—L929; (**E**) Resveratrol—L929; (**F**) Baicalin—L929.

**Table 1 pharmaceuticals-16-00082-t001:** Estimated Brian and Cousens model parameters and values of the maximum relative effects for MTT assays.

		Parameters					Goodness of Fit	
Cell Line	Substance	*b*	*e*	*f*	*x_op_* [mg/L]	*E_max_*	R^2^	RMSE
Balb3t	Curcumin	1.06	0.61	1.85	0.59	1.06	0.12	0.13
	Resveratrol	1.57	39.07	0.0042	0.76	1.00	0.25	0.15
	Baicalin	1.09	0.74	1.59	0.67	1.09	0.12	0.20
L929	Curcumin	1.66	6.24	0.25	3.55	1.35	0.27	0.34
	Resveratrol	3.53	38.98	0.00032	4.15	1.00	0.06	0.19
	Baicalin	1.12	1.00	1.16	0.67	1.08	0.31	0.15

**Table 2 pharmaceuticals-16-00082-t002:** Results of the *t* test. The hypothesis of equal means for samples and equal but unknown variances is: 0 = NOT rejected; 1 = rejected for significance level α = 0.05.

Cell Line	Substance	Dose [mg/L]	*p*-Value	Hypothesis
Balb3t3	Curcumin	0.5	0.073	0
		1	0.102	0
	Resveratrol	0.5	0.890	0
		1	0.921	0
	Baicalin	0.5	0.356	0
		1	0.002	1
L929	Curcumin	3	Not tested	
		6	0.009	1
	Resveratrol	3	0.708	0
		6	0.749	0
	Baicalin	0.5	Not tested	
		1	0.117	0

**Table 3 pharmaceuticals-16-00082-t003:** Estimated Brian and Cousens model parameters and values of maximum relative effects for ST tests.

		Parameters					Goodness of Fit	
Cell Line	Substance	*b*	*e*	*f*	*x_op_*	*E_max_*	R^2^	RMSE
Balb3t3	Curcumin	2.57	6.21	1.24	4.73	4.59	0.74	0.78
	Resveratrol	1.91	1.15	5.31	1.01	3.56	0.82	0.57
	Baicalin	1.53	1.69	6.44	2.28	6.07	0.87	0.60
L929	Curcumin	2.96	7.34	0.65	5.16	3.23	0.76	0.53
	Resveratrol	2.09	1.49	5.98	1.29	5.01	0.84	0.74
	Baicalin	1.43	1.68	4.82	2.57	4.72	0.81	0.54

## Data Availability

Data is contained within the article or Appendix A.

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
