# Peer review of "Supporting the Wound Healing Process—Curcumin, Resveratrol and Baicalin in In Vitro Wound Healing Studies"

_pharmaceuticals, 2023, doi:10.3390/ph16010082_

Round 1
Reviewer 1 Report
· Abstract is written at a high level of generality, please provide some specific values.
· There are several typos in the manuscript (e.g. unnecessary capital letters, double space, etc.).
· All abbreviations should be defined when used for the first time.
· Emphasise the novelty of the study.
· Give more detailed description on cell culturing (conditions, detaching, viability, etc.).
· Line 145: please give superscript for ‘4’.
· Section 1.1. How the stocks were prepared? What solvent was used?
· Line 162: CO2? Humidity?
· I'm curious how the Authors managed to make a scratch with a tip from a pipette? I've never managed to do it. It always damages the plastic in the culture dish and distorts the image. Besides, this scratch is always very wide, so that the field in the microscope does not include it. Such a scratch cannot be made even with a needle without damaging the plastic - the bottom of the vessel.
· Line 204: the section number is not correct, it should be ‘2’.
· Results and Discussion section should be divided into subsections with subtitles – this will make it easier to study and understand the results.
· What is the applicable value of the research, please discuss.
· Can tested compounds show cytotoxicity at higher concentrations that you have not tested?
Author Response
There are several typos in the manuscript (e.g. unnecessary capital letters, double space, etc.).
Thank You for Your suggestion. The noticed typos were corrected.
- All abbreviations should be defined when used for the first time.
Thank You for Your suggestion. The missing explanations of abbreviations were added.
- Emphasise the novelty of the study.
So far, no studies have been published that would use the Brian and Cousens model to describe the mitochondrial or cellular response to the applied doses of bioflavonoids. The results obtained can be helpful in developing a modern healing dressing based on natural medicinal products.
- Give more detailed description on cell culturing (conditions, detaching, viability, etc.).
Thank You for Your suggestion. In manuscript we updated cell culturing details as follows:
Presented study was performed on L929 and Balb3t3 cell lines. Cell lines were obtained from Sigma-Aldrich (St.Louis, MO, USA). Both cell lines were cultured at 37 °C, 5% CO2, relative humidity was set at 95%. Cell lines were passaged twice a week, and the experiments were performed on cells between passage 15 and 25. The culture medium was changed every 48 h. Every dictating was performed by presented procedure: after removing the supernatant from L929 and Balb3t3 cells, they were washed with PBS (Phosphate Buffered Saline), then treated with TrypLE Express solution (Gibco, Thermo Fisher Scientific, Waltham, MA, USA) and incubated for 5 min at 37 °C. Next, the cells were transferred to a centrifuge tube, complete medium was added (to inactivate the TrypLE Express), and cells were centrifuged for 5 min at 1000 g. The supernatant was removed, and cells were resuspended in fresh medium. Viability of a cell line was every day check by microscope observation of cell culture morphology.
- Line 145: please give superscript for ‘4’.
Mistake was taken during moving manuscript form original file into template (104 instead of 104). Thank You for noticing it, the mistake was corrected.
- Section 1.1. How the stocks were prepared? What solvent was used?
The test stocks were prepared by the following procedure:
to the bioflavonoids, DMSO was added to the final concentration set at 10mM. The stock solutions were stored at -80C. paying attention that the storage time does not exceed 6 months. Every time before the planned biological assay the stock solutions were dissolved with proper culture media. Used culture media wa\ere with reduced of the FBS level - till 5%. During the test, crucial was to set the level of DMSO concentration lower than 1% in the highest test bioflavonoid concentration.
- Line 162: CO2? Humidity?
Thank You for Your suggestion. We added those information: 5% CO2, relative humidity 95%.
- I'm curious how the Authors managed to make a scratch with a tip from a pipette? I've never managed to do it. It always damages the plastic in the culture dish and distorts the image. Besides, this scratch is always very wide, so that the field in the microscope does not include it. Such a scratch cannot be made even with a needle without damaging the plastic - the bottom of the vessel.
For years, in our laboratory, the scratch test has been performed by removing part of the cells from the bottom of the vessel using a pipette tip. During many years of research, the plastic was never scratched when the pipette tip was gently and firmly moved across the bottom of the vessel. The pipette tip is guided at an angle of 30 degrees to the bottom of the vessel. This positioning of the pipette tip to the bottom allows for delicate and reliable creation of a culture defect with a width that can be measured in a standard cell culture microscope. Each scratch is made once. The researcher does not correct the resulting loss so as not to distort the obtained image.
- Line 204: the section number is not correct, it should be ‘2’.
Mistake was taken during moving manuscript form original file into template. Thank You for noticing it, the mistake was corrected.
- Results and Discussion section should be divided into subsections with subtitles – this will make it easier to study and understand the results.
Thank You for Your suggestion. Results and Discussion section has been divided into subsections.
- What is the applicable value of the research, please discuss.
The approach used by the authors allows to find the most effective concentration of the active substance for which the fastest division of fibroblasts occurs, which can directly translate into shorter wound healing time. The research proposed by the researchers may translate into the creation of a new type of dressings supporting the healing of hard-to-heal wounds (e.g. bedsores). In addition, they can be an indication for further research on the combination of active ingredients in order to create a product with the most effective effect.
- Can tested compounds show cytotoxicity at higher concentrations that you have not tested?
Than You for this question. Based on experience and available knowledge and literature data, it was assumed that the tested substances will have a toxic effect at higher concentrations. The purpose of the study was to check the pro-proliferative properties of the tested substances. It is known that resveratrol will inhibit the culture when its concentration exceeds the appropriate level (this is used in cancer research). In the project, it was decided to use the presented concentration values.
Reviewer 2 Report
The article by Kacper Jagiełłoa et al. it has been checked. This manuscript contains a piece of the interesting working effect of different bioflavonoids, curcumin, resveratrol and baicalin, on the wound healing process using an in vitro model as a reference. For this, a cytotoxicity assay was performed using an MTT assay to assess the toxicity of the compounds. Then, the Scratch test (ST) was obtained to evaluate the influence of these compounds on the healing process. As a result, the bioflavonoids studied showed a simulating effect on the proliferative capacity of L929 and Balb3t3 cell lines. The paper is well-written and the results are of interest for the scientific community. However, some minor points should be considered before paper publication in Pharmaceuticals.
-Study concept and design of the manuscript should be more clearly described in the introduction section of the paper.
-The reason for selecting L929 and Balb3t3 cell lines for MTT assay should be explained and included in the main text.
-The results of the t-test for selected flavonoids at concentrations 3, 6, 9, 12, 148 15, 20, 25 μg/mL must be included in table 2 of the manuscript.
-The absorbance and fluorescence spectra of curcumin is displayed in the same wavelength region that the spectra of MTT. How could this fact influence the values obtained for viability?. A method for viabilities correction can be found in the following reference:Giráldez-Pérez, R.M.; Grueso, E.; Montero-Hidalgo, A.J.; Luque, R.M.; Carnerero, J.M.; Kuliszewska, E.; Prado-Gotor, R. Gold Nanosystems Covered with Doxorubicin/DNA Complexes: A Therapeutic Target for Prostate and Liver Cancer. Int. J. Mol. Sci. 2022, 23, 15575. https://doi.org/10.3390/ijms232415575.
Author Response
In this study, the effects of selected bioflavonoids on the in vitro wound healing process were examined. The influence of different concentrations of curcumin, resveratrol, and baicalin on cell viability (MTT Assay) and cell migration (Scratch Assay) was evaluated. The use of mathematical modeling is designed to determine the most effective concentrations of selected substances that support the fibroblast proliferation process. So far, no studies have been published that would use the Brian and Cousens model to describe the mitochondrial or cellular response to applied doses of bioflavonoids. The results obtained can be helpful in developing a modern healing dressing based on natural medicinal products.
-The reason for selecting L929 and Balb3t3 cell lines for MTT assay should be explained and included in the main text.
The cell lines used in the study are well characterized. The tests presented are well designed for the lines used. These information had an impact on the selection of the cells used; however, the main reason for the choice of fibroblasts was the following facts: In the in vivo system, fibroblasts are one of the foundations of the healing process. These cells multiply and create a matrix for the developing tissue. It is the fibroblasts that are involved in scar formation. It seems natural to use them in the study of substances that are supposed to have a wound-healing effect. Supporting the proliferation of fibroblasts in the wound will have an impact on the improvement in the healing process. Further studies are planned using cell lines that are the models of such generations of cells in the wound as epithelial cells, keratinocytes, vascular cells, etc.
The cell lines used in the study are well characterized. Furthermore, fibroblasts are one of the foundations of the healing process. These cells multiply and create a matrix for the developing tissue. It is the fibroblasts that are involved in scar formation. It seems natural to use them in the study of substances that are supposed to have a wound-healing effect. Supporting the proliferation of fibroblasts in the wound will have an impact on the improvement in the healing process.
-The results of the t-test for selected flavonoids at concentrations 3, 6, 9, 12, 15, 20, 25 μg/mL must be included in table 2 of the manuscript.
The results of the t-test for other concentrations mentioned in table 2 will not give any additional information. The t-test was performed to answer the question – Is there a statistical difference between the control sample and samples where stimulation of viability was probable? We were looking for higher viability in the regions shown by the Brian and Cousens model. The results of the t-test prepared for higher concentration can give information about probable toxic effects. The results of t-test for higher connotations are presented in Supporting Materials.
-The absorbance and fluorescence spectra of curcumin is displayed in the same wavelength region that the spectra of MTT. How could this fact influence the values obtained for viability?. A method for viabilities correction can be found in the following reference:Giráldez-Pérez, R.M.; Grueso, E.; Montero-Hidalgo, A.J.; Luque, R.M.; Carnerero, J.M.; Kuliszewska, E.; Prado-Gotor, R. Gold Nanosystems Covered with Doxorubicin/DNA Complexes: A Therapeutic Target for Prostate and Liver Cancer. Int. J. Mol. Sci. 2022, 23, 15575. https://doi.org/10.3390/ijms232415575.
The absorbance of MTT was measured as an absorbance above that in blank samples where the curcumin (and others) were present. There's no need to use a posteriori correction.
Round 2
Reviewer 1 Report
I have no more comments.